# Exploring the Expression of Cardiac Regulators in a Vertebrate Extremophile: The Cichlid Fish *Oreochromis (Alcolapia) alcalica*

**DOI:** 10.3390/jdb8040022

**Published:** 2020-10-04

**Authors:** Gemma Sutton, Lewis J. White, Antonia G.P. Ford, Asilatu Shechonge, Julia J. Day, Kanchon K. Dasmahapatra, Mary E. Pownall

**Affiliations:** 1Biology Department, University of York, York, YO10 5DD, UK; gs569@exeter.ac.uk (G.S.); ljw569@york.ac.uk (L.J.W.); kanchon.dasmahapatra@york.ac.uk (K.K.D.); 2Department of Life Sciences, Centre for Research in Ecology, Whitelands College, University of Roehampton, Holybourne Avenue, London, SW15 4JD, UK; Antonia.Ford@roehampton.ac.uk; 3Tanzania Fisheries Research Institute, P.O.BOX 98 Kyela, Mbeya, Tanzania; ashechonge@yahoo.com; 4Department of Genetics, Evolution and Environment, University College London, Darwin Building, Gower Street, London, WC1E 6BT, UK; j.day@ucl.ac.uk

**Keywords:** cichlid fish, extremophile, environmental adaptation, cardiac myogenesis

## Abstract

Although it is widely accepted that the cellular and molecular mechanisms of vertebrate cardiac development are evolutionarily conserved, this is on the basis of data from only a few model organisms suited to laboratory studies. Here, we investigate gene expression during cardiac development in the extremophile, non-model fish species, *Oreochromis (Alcolapia) alcalica*. We first characterise the early development of *O. alcalica* and observe extensive vascularisation across the yolk prior to hatching. We further investigate heart development by identifying and cloning *O. alcalica* orthologues of conserved cardiac transcription factors *gata4*, *tbx5*, and *mef2c* for analysis by in situ hybridisation. Expression of these three key cardiac developmental regulators also reveals other aspects of *O. alcalica* development, as these genes are expressed in developing blood, limb, eyes, and muscle, as well as the heart. Our data support the notion that *O. alcalica* is a direct-developing vertebrate that shares the highly conserved molecular regulation of the vertebrate body plan. However, the expression of *gata4* in *O. alcalica* reveals interesting differences in the development of the circulatory system distinct from that of the well-studied zebrafish. Understanding the development of *O. alcalica* embryos is an important step towards providing a model for future research into the adaptation to extreme conditions; this is particularly relevant given that anthropogenic-driven climate change will likely result in more freshwater organisms being exposed to less favourable conditions.

## 1. Introduction

Aquatic ectotherms, such as fish, are significantly impacted by the changes or stressors in their environment, as they rely on the surrounding water for maintaining homeostasis [1,2]. Despite this restriction, fish have evolved morphological, biochemical, physiological, behavioural, and developmental mechanisms allowing them to colonise nearly all aquatic environments, including a variety of extreme habitats [3,4]. Although most fish species are unable to survive extremes of temperature, pH, salinity, and environments which seasonally dry out, some specialised species have adapted to thrive in such conditions, making them useful subjects for studying the molecular and developmental mechanisms underpinning adaptation to imposed stressors [5]. Exploring the naturally evolved adaptations in extremophile vertebrates will give insight into core biological functions, which may provide important information for future research in vertebrate biology, as well as the diversity in animal development, and it may even contribute to our understanding of human disease [6,7,8].

Knowledge of development is key for understanding mechanisms of evolutionary variation, as changes in developmental processes can result in novel phenotypes [9]. Currently, the majority of knowledge of early ontogeny of fishes stems from studies in the model organisms zebrafish (*Danio rerio*) and medaka (*Oryzias latipes*) [10,11]. These species were selected as suitable model organisms for developmental biology due to their size, short development periods, accessible genetic tools, and ability to be bred and reared easily in laboratory conditions. However, these selected teleosts may not be representative of early development in other fish species. The study of development in non-model species has the potential to provide additional insights into the molecular evolution of biological diversity in teleosts which are by far the most species-rich vertebrate clade. The Teleostei comprise ca. 34,000 species, accounting for nearly 98% of all actinopterygian (ray-finned) fishes, i.e., half of all extant vertebrates [12]. Research using non-model organisms will allow wider species comparisons and a better understanding of adaptive traits [13].

With their high species richness and recent adaptive radiations, cichlid fishes are a model system in evolutionary biology, particularly for studying speciation [14,15,16]. One subgenus, *Alcolapia* (nested within the genus *Oreochromis* [17]), presents a unique radiation of extremophile cichlids, inhabiting the extreme waters of the East African soda lakes of Natron and Magadi. This adaptive radiation comprises four described species, *Oreochromis (Alcolapia) alcalica, O. (A.) latilabris*, *O. (A.) ndalalani* (Lake Natron, Tanzania), and *O. (A.) grahami* (Lake Magadi, Kenya), and diverged as recently as 10,000 years ago from freshwater ancestors [18,19,20]. The *Alcolapia* fish experience water temperatures of 30–42 °C, pH 9–11.5, fluctuating dissolved oxygen levels (0.08–6.46 mg/L), and high salt concentrations (>20 ppt) [21]. Living in this environment has led to a number of key adaptations including 100% ureotelism, facultative airbreathing, a specialised gut morphology, and maintaining a heightened metabolic rate [22,23,24,25,26,27]

An aspect of development that shows striking conservation across animals with very different lifestyles, from insects to fish to tetrapods, is that of heart function to pump fluid around the body, transporting nutrients, metabolites, and oxygen to tissues [28,29,30]. How this remarkable adaptation develops during embryogenesis has been studied in several experimental model organisms, providing an understanding of conserved mechanisms of cardiac development in vertebrates [28]. As the primary teleost model organism, zebrafish has emerged as an important vertebrate model for studying cardiovascular development and disease [31].

Cardiac myogenesis in zebrafish begins with the emergence of cardiac progenitor cells at 5 h post fertilisation (hpf) in the lateral marginal zone of the blastula, with ventricular progenitors situated more dorsally and closer to the margin than atrial progenitors [32,33]. During gastrulation, the cardiac progenitors migrate to the anterior lateral plate mesoderm where bilateral progenitors fuse into the cardiac disc with endocardial cells in the centre, surrounded by ventricular cardiomyocytes, surrounded by atrial cardiomyocytes that elongate into the linear heart tube. By 24 hpf, heart tube contraction initiates [34,35] and the linear heart tube undergoes cardiac jogging, whereby left-right symmetry is broken as the heart tube migrates leftwards and begins looping [36]. At 48 hpf, the two-chambered heart is clearly distinguishable by the constriction of the atrioventricular canal [37].

The network of cardiac transcription factors that regulate embryonic heart development displays a high degree of evolutionary conservation across vertebrates; here, we identify and describe the expression of a subset of these regulators, GATA-binding protein 4 (GATA4), T-box 5 (Tbx5), and Myocyte enhancer factor 2c (Mef2c) in *Oreochromis (Alcolapia) alcalica*. Members of the GATA family of zinc-finger transcription factors are involved in the early specification of cardiac progenitors. The GATA factors are crucial during haematopoiesis and cardiac myogenesis in vertebrates and *Drosophila* [38,39]. Tbx5 is a member of the T-box family of transcription factors and is expressed during development of heart, eyes, and forelimbs. Mutations in *TBX5* in humans cause Holt–Oram syndrome (HOS), an autosomal-dominant disorder characterised by forelimb malformations and cardiac defects [40,41,42]. Cardiac abnormalities of HOS patients include septation defects and conduction disease [43]. Myocyte enhancer factor 2 (Mef2) proteins are MADs (MCM1, Agamous, Deficiens, Serum response factor)-box transcription factors. There are four vertebrate *mef2* genes, *mef2a*–*d*, expressed in precursors of heart, skeletal, and smooth muscle lineages [44]. Loss of function of the single *mef2* gene in *Drosophila* results in abnormal development of all muscle types, including cardiomyocytes [45]. In mice, *Mef2a, c*, and *d* are expressed in cardiac mesoderm and *Mef2a*- or *Mef2c*-null mice display cardiac developmental defects [46].

We report here methods to acquire and culture *Oreochromis (Alcolapia) alcalica* embryos and to interrogate gene expression patterns using whole-mount in situ hybridisation. We present the first developmental stage series of *O. alcalica* and the expression patterns of the *O. alcalica* orthologues of the cardiac regulatory genes *gata4*, *tbx5*, and *mef2c* as assessed by in situ hybridisation.

## 2. Methods

Live specimens of *O. alcalica* were collected from a single spring (site 5) [20,47] at Lake Natron in Tanzania (permit 2017-259- NA-2011-182) in June 2017. Fish were packaged individually in breather bags and transported to Bangor University to establish a breeding colony. Some of the *O. alcalica* specimens were subsequently moved to the University of York and housed in a recirculating aquarium (Aquatics Habitat) with constant water conditions kept at a temperature of 30 °C, pH 9, and conductivity of 3800 µS. A separate zebrafish system was maintained at 27 °C, pH 7.4, and conductivity 800 µS. This study was carried out using procedures authorised by the United Kingdom (UK) Home Office in accordance with the Animals Scientific Procedures Act (1986) and approved by the Animal Welfare and Ethical Review Body at the University of York and the UK Home Office project licence to MEP (POF245295).

### 2.1. Fish maintenance and Embryo Collection

To control *O. alcalica* breeding, dividers were placed in tanks to separate a single male from a group of 3–4 females. After acclimatising for several days, dividers were removed and mating behaviour proceeded. *O. alcalica* is a mouth-brooding cichlid, in which females carrying fertilised eggs are identified by an enlarged buccal cavity. To obtain embryos, the females were removed from the water, and embryos were released by applying light pressure to the buccal cavity. After collection, *O. alcalica* embryos were incubated at 29–30 °C gently shaking in system water with fungicide methylene blue (0.0003 mg/mL) (Sigma). Zebrafish (AB) embryos were collected as per standard procedures. At specific stages, *O. alcalica* and zebrafish embryos were dechorionated and fixed for 1 h at room temperature in MEMFA (0.1 M 3-(*N*-Morpholino)propanesulfonic acid (MOPS), pH 7.4, 2 mM Ethylene glycol tetraacetic acid (EGTA), 1 mM MgSO_4_, 3.7% formaldehyde) and stored at −20 °C in 100% methanol.

### 2.2. Phylogeny

Zebrafish *gata4*, *tbx5*, and *mef2c* coding sequences were identified on ZFIN (https://zfin.org/). BLAST (https://blast.ncbi.nlm.nih.gov/Blast.cgi) was used to identify these sequences in the published genome of the freshwater cichlid *Oreochromis niloticus*. *Oreochromis niloticus* sequences were aligned to the unpublished *O. alcalica* genome (Dashamapatra, in preparation) to identify *O. alcalica* coding sequences of the cardiac regulators; we found these putative orthologues to have 60% (Tbx5), 63% (Mef2c), and 62% (GATA4) amino-acid sequence identity with the zebrafish proteins. The *gata4*, *tbx5*, and *mef2c* sequences were retrieved from *Drosophila melanogaster*, *Takifugu rubripes*, *Maylandia zebra*, *Nothobranchus furzeri*, *Oryzias latipes*, *Danio rerio*, *Callorhinchus milii*, *Xenopus tropicalis*, *Gallus gallus*, *Mus musculus*, *Pan troglodytes*, and *Homo sapiens* using BLAST. The predicted full-length amino-acid sequence was deduced from the coding sequence of each gene, and multiple sequence alignment was conducted using MEGA-X by MUSCLE (Robert C. Edgar, 2004; codon alignment). Phylogenetic tree reconstruction was conducted in MEGA-X on individual genes using maximum likelihood [48]. The tree with the highest log likelihood is presented in Figures 2–4. The percentage of trees in which the associated taxa clustered together is shown next to the branches. Initial trees for the heuristic search were obtained automatically by applying Neighbour-Join and BioNJ algorithms to a matrix of pairwise distances estimated using a Jones-Taylor-Thornton (JTT) model, and then selecting the topology with a superior log likelihood value. A discrete Gamma distribution was used to model evolutionary rate differences among sites, and a tree was drawn to scale, with branch lengths measured in the number of substitutions per site.

### 2.3. RNA Extraction, Complementary DNA (cDNA) Synthesis, and Reverse-Transcription PCR (RT-PCR)

RNA was extracted from *O. alcalica* and zebrafish embryos using TRI Reagent (Sigma-Aldrich) using phase separation and isopropanol precipitation. Here, 1 µg of total RNA was used for cDNA synthesis with Superscript IV (Invitrogen) and random hexamers (Thermo Scientific). Primers (Sigma) for the amplification of *O. alcalica* cardiac regulatory genes were designed against sequences identified in the *O. alcalica* genome using PrimerSelect (DNASTAR) (Table 1) to amplify regions of 400–600 bp in length; a similar approach was used for zebrafish orthologues using sequences in NCBI. The cDNAs were amplified by RT-PCR, cloned into pGEM T-Easy (Promega), and sequenced. The following sequences were submitted to Genbank: *Oa tbx5* (MT904199), *Oa mef2c* (MT904200), and *Oa gata4* (MT904201). For generating antisense RNA probes, SalI (Promega) linearised plasmids were used as templates for T7 transcription (Ambion), and NcoI (Promega) linearised plasmids were used as templates for SP6 transcription (Ambion), depending on orientation of the insert. In vitro run-off transcription at 37 °C was used to incorporate digoxigenin (DIG)-labelled UTP analogue (Roche).

### 2.4. Whole-Mount In Situ Hybridisation

Fixed embryos were rehydrated with washes of decreasing levels of methanol/phosphate-buffered saline-Tween-20 (PBSAT) (75% methanol/PBSAT, 50% methanol/PBSAT, 100% PBSAT) and treated with 10 µg/mg proteinase K (Roche) at room temperature. Zebrafish embryos were treated with proteinase K for 10 min per day post fertilisation (dpf). *O. alcalica* embryos of 2dpf, 3dpf, and 7dpf were treated with proteinase K for 4, 15, and 30 min, respectively. Embryos were treated with 0.1 M triethanolamine/acetic anhydride and washed in PBSAT. Following post-fixation with 10% formalin and 2 h incubation at 68 °C in hybridisation buffer (50% formamide (Ambion), 1 mg/mL total yeast RNA, 5× saline-sodium citrate (SSC), 100 µg/mL heparin, 1× Denharts, 0.1% Tween-20, 0.1% CHAPS, 10 mM EDTA), embryos were incubated in 1 mL of hybridisation buffer with 3–6 µL of DIG-labelled antisense probe at 68 °C overnight.

Embryos were extensively washed at 68 °C in 2× SSC + 0.1% Tween-20, 0.2× SSC + 0.1% Tween-20, and maleic acid buffer (MAB; 100 mM maleic acid, 150 mM NaCl, 0.1% Tween-20, pH 7.8). This was replaced with pre-incubation buffer (4× MAB, 10% BMB, 20% heat-treated lamb serum) for 2 h. Embryos were incubated overnight (rolling at 4 °C) with fresh pre-incubation buffer and 1/2000 dilution of anti-DIG coupled with alkaline phosphatase (AP) (Roche). Embryos were washed in 1× MAB and AP buffer (100 mM Tris, 50 mM MgCl, 100 mM NaCl, 0.1% Tween-20). BM purple (Roche), the substrate for AP, was applied to embryos and left at room temperature until colour developed. Embryos were fixed in MEMFA and photographed. The 7dpf *gata4* embryos were washed in PBSAT and bleached in 5% H_2_O_2_ (Merck) under bright light to remove pigmentation. Once bleached, embryos were washed in PBSAT and stored in MEMFA. Stack images of the in situ hybridisations were taken using a SPOT Camera (14.2 Colour Mosaic, Diagnostic Instruments). Focus stacking and image editing was undertaken on Adobe Photoshop 2020.

## 3. Results

### 3.1. Early Development of O. alcalica

The early development of *O. alcalica* during the first 10 dpf at 28 °C is documented in Figure 1. Embryonic development could be subdivided into six periods: zygote (Figure 1A), cleavage (Figure 1B), blastula-gastrula (Figure 1C,D), segmentation (Figure 1E), pharyngula (Figure 1F–H), and hatching (Figure 1G). *Oreochromis (Alcolapia) alcalica*’s embryonic development is similar to that of other mouth-brooding cichlids [49,50,51]. The embryo was surrounded by the chorion, a transparent membrane that stuck closely to the egg which persisted until hatching (Figure 1A–H). The yolk was opaque yellow and homogeneous in appearance, making observations of the embryonic anlage difficult. At early embryonic stages, there was almost no perivitelline space between the chorion and yolk, and dechorionation was difficult without puncturing the yolk (Figure 1A–D).

Newly fertilised eggs of *O. alcalica* had an ovoid shape, with the animal pole narrower than the vegetal pole (Figure 1A). The blastodisc sat at the animal pole. The first cleavage furrow was meridional and occurred at 2 hpf, resulting in two blastomeres (Figure 2B). The further mitotic divisions that occur during the cleavage period resulting in many blastomeres were not distinguishable under a dissecting microscope. Furthermore, the specific timing and cellular dynamics of *O. alcalica* gastrulation were difficult to observe due to the yolk (Figure 1C,D). Epiboly was previously described in Nile tilapia, *Oreochromis niloticus*, another mouth-brooding cichlid that is closely related to *O. alcalica* [17], by sequential fixation of blastula-gastrula stage embryos [49]. As *O. alcalica* embryos dissociated rapidly when dechorionated and fixed before the segmentation stage, we were unable to document gastrulation in *O. alcalica* in this study.

By 42 hpf, the *O. alcalica* embryos completed gastrulation and entered segmentation period (Figure 1E). At 66 hpf, the embryos entered the pharyngula period. Melanocytes emerged from the embryonic axis and began migrating across the yolk. Embryos had a clearly formed head with unpigmented eyes (Figure 1F). At this stage, a linear heart tube formed, and rhythmic contractions could be observed (Appendix A). At 3 dpf, the head developed upwards from the yolk. Early vasculature development was observed in patches on the yolk and around the heart (Figure 1G). This early vasculature was pumped across the yolk, through the heart tube and embryo (Appendix A). By 4 dpf, the eyes developed pigmentation, the heart started to loop, and extensive vasculature developed across the yolk (Figure 1H, Appendix A). At 5 dpf, embryos hatched from the chorion membrane (Figure 1I). At 5–6 dpf, the heart looping was complete and the two-chambered heart formed (Figure 1I–J, Appendix A). After hatching, many aspects of the adult body plan were apparent, and embryos could swim actively by 7 dpf (Figure 1J–M).

*O. alcalica* rapidly develops adult morphology without a prolonged, free-feeding larval stage [49,51] and, therefore, fits the definition of a direct-developing fish species which have large, yolk-rich eggs and complete their development while living off the maternally deposited yolk supply, until transforming directly into a free-feeding juvenile [52,53].

One of the most striking aspects of *O. alcalica* development was the emergence of an extensive vascular system which began in patches on the yolk and subsequently branched and formed a network of vessels across the yolk between 3 and 4 dpf (Figure 1G,H). In the closely related freshwater Nile tilapia (*O. niloticus*), the earliest heartbeat and blood circulation were reported at 40–42 hpf, and yolk vascularisation was also observed at 4 dpf [49,54]. When cultured at about the same temperature used in these previous studies (28–30 °C), we found and recorded heart contractions at 66 hpf, 3 dpf, 4 dpf, and 5 dpf, as shown in movies in the Appendix A. The early vascularisation seen in these cichlids is very unlike that of *Danio rerio* and shows more similarity to vascularisation seen in aminotes such as chick embryos. To investigate this further, the genes for the conserved cardiac transcription factors, *gata4*, *tbx5*, and *mef2c*, were cloned from extracted RNA and used for in situ hybridisation studies in *O. alcalica*. The expression of these genes was previously described in zebrafish [55,56,57]; nevertheless, they are included here for comparison. To do this, *gata4*, *tbx5*, and *mef2c* were cloned from zebrafish and analysed by in situ hybridisation, as presented in Appendix A).

### 3.2. gata4 is Expressed in Cardiac and Haematopoietic Regions of O. alcalica Embryos

To confirm that the cDNA isolated from *O. alcalica* was indeed *gata4*, a phylogenetic analysis was undertaken and revealed high conservation of our sequence with known GATA4 proteins (Figure 2A). In situ hybridisation analysis of embryos fixed at 2 dpf showed that the expression of *Oa gata4* in the linear heart tube was localised at the anterior left side of the embryo (Figure 2B,C), consistent with our observations of the early pharyngula stage, where the contracting linear heart tube forms (Figure 1F, Appendix A). The *gata4* expression in the heart tube was more restricted by 3 dpf (Figure 2E,F), and, by 7 dpf, *gata4* transcripts were still detected in a limited region of the heart. The *gata4* in situ hybridisation of zebrafish at 1 dpf identified *gata4*-expressing cardiomyocytes in the anterior lateral plate mesoderm (Appendix A), and, by 2 dpf and 3 dpf, *gata4* transcripts were expressed in the anterior midline (Appendix A). Other regions of *gata4* expression were identified at the posterior of the embryo in the haematopoietic region (Figure 2B,C).

Later in development, at 3 dpf and 7 dpf, *gata4* transcripts localised to numerous puncta across the yolk (Figure 2D–H). The embryos in Figure 2F–H were bleached to remove pigmentation from melanocytes, allowing better visualisation of these puncta. The *gata4* expression on the yolk, overlapping with the extensive vasculature, was first distinguishable at 3 dpf (Figure 1F). These extraembryonic *gata4*-expressing puncta were likely blood islands (Figure 2D–H), a feature of haematopoiesis that is absent in zebrafish, but present during amniote development.

As *O. alcalica* are direct-developers, they are more dependent on the yolk as an energy source than indirect-developers that have a free-feeding larval stage such as zebrafish. The development of this early vasculature from blood islands likely provides a vital, early energy supply in *O. alcalica* embryos. Interestingly, it is not *GATA4*, but *GATA1* and *GATA2* that are expressed in the blood islands of chicks and mice [59,60].

### 3.3. tbx5 is Expressed in the Developing Pectoral Fin, Eyes, and Heart of O. alcalica

The expression of *Oa tbx5* was found in the heart, limb, and eye by in situ hybridisation, consistent with its expression in other vertebrate embryos. Phylogenetic analysis of Tbx5 protein in multiple vertebrates confirmed the identity of the *tbx5* cDNA isolated from *O. alcalica* (Figure 3A). At 2 dpf and 3 dpf, *Oa tbx5* transcripts were detected and localised specifically to the dorsal region of the eyes (Figure 3B,C). This expression in the eye was easily visualised prior to pigmentation of the retina in *O. alcalica*; however, by 7dpf, *Oa tbx5* expression in the eye was not visible in the retinal pigment epithelium (RPE) due to the black pigmentation (Figure 3F,G). This expression of *Oa tbx5* in the dorsal retina is conserved in vertebrates [61,62,63], and, in zebrafish, *tbx5b* was expressed in the dorsal retina from 1–3 dpf (Appendix A).

At 3 dpf, there was transient expression of *Oa tbx5* in the heart (Figure 3D), as also seen in zebrafish (Appendix A). The expression of *Oa tbx5* in *O. alcalica* hearts at 3 dpf, during the linear heart tube stage, is consistent with the known role of Tbx5 in regulating cardiac looping and jogging in zebrafish. At 3 dpf, *Oa tbx5* transcripts localised to the *O. alcalica* pectoral limb bud (LB) (Figure 3D,E). *Oa tbx5* was still expressed in the LB at 7 dpf (Figure 3F, G). Similarly, in zebrafish, *tbx5b* was expressed in the LB from 1–3 dpf (Appendix A).

### 3.4. mef2c is Expressed in Developing Muscle in O. alcalica

The expression of the highly conserved cardiomyogenic regulator, *mef2c*, was assessed in *O. alcalica* by in situ hybridisation. In *Drosophila*, *D-mef2* is essential for the formation of cardiac muscle, and loss of *D-mef2* results in failure of cardiomyocyte differentiation [45,65]. In zebrafish, Mef2c is also required for cardiac development; combinatorial loss-of-function of *mef2ca* and *mef2cb* causes developmental arrest of cardiomyocytes. These *mef2c* paralogues control the expression of myocardial sarcomeric genes [66]. Phylogenetic analysis of Mef2c protein in multiple vertebrates confirmed the identity of the *Oa mef2c* cDNA isolated from *O. alcalica* (Figure 4A), and it was distinct from the highly related Mef2d sequences (Appendix A). Consistent with previous studies, *Dr mef2ca* transcripts were detected by in situ hybridisation in early cardiac regions in zebrafish (Appendix A); however, we found no expression of *Oa mef2c* in cardiac regions of *O. alcalica*. The expression shown in Figure 4B–E is similar to that described in *Xenopus laevis* [67], and we expect that the stages we analysed were too late to detect *Oa mef2c* in the heart because it is an early regulator of cardiac differentiation. Nonetheless, our expression analysis provides insight into the process of myogenesis in *O. alcalica;* at 2 dpf and 3 dpf, *Oa mef2c* was robustly expressed in the somites (Figure 4B–E), the embryonic source of all skeletal muscle in the vertebrate body [68]. This expression is conserved in zebrafish where *mef2ca* transcripts were also detected in somites from 1–3 dpf (Appendix A).

At 7 dpf, *Oa mef2c* continued to be expressed in the distinctive chevron structure of the somites and in hypaxial muscle. These *Oa mef2c*-expressing streams of migratory muscle precursors (MMPs) migrated to form the muscles of the ventral body wall (arrowheads in Figure 4F,G). In *O. alcalica*, *Oa mef2c* expression revealed that the MMPs spread out as direct extensions of anterior somites that extended ventrally across the yolk; these extensions were not seen in zebrafish embryos.

## 4. Discussion

We report here the expression of three evolutionarily conserved regulators of cardiogenesis in the context of a direct-developing extremophile fish, *Oreochromis (Alcolapia) alcalica.* We include a developmental stage series and a set of videos to illustrate the onset of heart development and contraction.

### 4.1. gata4

Bilateral cardiomyocytes in the anterior lateral plate mesoderm can be identified through expression of *gata4/5/6* [69,70]. GATA4, in particular, is a potent driver of cardiac myogenesis; ectopically expressed *gata4* induces pluripotent *Xenopus* animal cap organoids to form contracting cardiomyocyte tissue [71]. Furthermore, human congenital heart defects are linked to mutations in *GATA4*, including valve and septal defects [72,73]. Morpholino oligonucleotide (MO) knockdown of *gata4/5/6* in *Xenopus* and zebrafish embryos attests to their roles in cardiac myogenesis. *Xenopus* morphants of individual GATA factors display cardia bifida, whereas knockdown of all three GATA factors eliminated expression of markers of cardiac differentiation [74]. In zebrafish, however, MO knockdown of *gata5* and *gata6* resulted in cardia bifida and substantial reduction in expression of contractile protein genes, whereas *gata4* morphant cardiomyocytes migrated to the midline normally [74]. The difference in phenotype severity of *gata5* morphants in *Xenopus* and zebrafish suggests a change in the activity of these GATA factors during evolution [75].

Our finding that *Oa gata4* is expressed in blood islands, the bipotential precursors of both vascular endothelia and blood cells, is interesting. In zebrafish and mammals, haematopoiesis occurs in shifting anatomical regions of the developing embryo [76]. In zebrafish, haematopoietic stem cells (HSCs) arise from the hemogenic endothelium lining the ventral wall of the dorsal aorta (VDA) [77]. HSCs originating from the VDA seed three haematopoietic organs: the caudal haematopoietic tissue (CHT), the thymus, and the kidney, the adult site of haematopoiesis in zebrafish [78]. In mammals, the VDA is functionally equivalent to the aorta-gonad-mesonephros (AGM) [79,80]. Mammalian HSCs emerge from the AGM, and transiently expand in the foetal liver before being maintained in the bone marrow, the adult site of haematopoiesis [81,82,83]. The mammalian foetal liver is equivalent to the zebrafish CHT, a vascularised region in the tail [84]. It was shown that primary cell lines generated from the zebrafish CHT at 3 dpf support HSC proliferation and differentiation [85]. The posterior region expressing *Oa gata4* in *O. alcalica*, labelled as the haematopoietic region (Figure 2B,C), may be similar to the CHT in zebrafish.

Blood islands in mice and chick form as clusters of mesodermal cells in the yolk sac that accumulate haemoglobin and are surrounded by outer endodermal cells that flatten and later form the endothelium of blood vessels [86]. The mesodermal and endodermal lineages that form the blood islands derive from common precursors known as hemangioblasts [87]. Zebrafish do not form yolk sac blood islands during embryogenesis. However, in some fish, such as killifish (*Fundulus spp*.), angelfish (*Pterophylum scalare*), and chondrichthyans, blood development predominantly occurs in the yolk sac [88]. This suggests that there is diversity in the location of embryonic blood development in teleosts.

The sites of haematopoiesis and vasculogenesis in *O. alcalica* embryos are more aligned with blood development in amniotes than with zebrafish. In vertebrates, the conserved GATA factors are divided into two subfamilies; *gata1/2/3* are expressed in developing blood cells, and *gata4/5/6* are expressed in mesodermal and endodermal tissues such as heart, liver, pancreas, and gut [89]. In the murine yolk sac, *GATA1* and *GATA2* transcripts were detected in blood islands by in situ hybridisation. Furthermore, *GATA1-* and *GATA2*-null mouse embryos are deficient in erythroid cells [60,90,91,92]. This suggests that, in mice, *GATA1* and *GATA2* are crucial for haematopoietic development in the yolk sac. Our novel finding that *Oa gata4* transcripts localise to blood islands of *O. alcalica* embryos raises the possibility of the neofunctionalisation of *Alcolapia* GATA factors. Therefore, it would be valuable to undertake further research with the aim of characterising the expression and function of other GATA factors in *O. alcalica* development.

### 4.2. tbx5

Due to its clinical relevance, *tbx5* loss-of-function vertebrate models have been developed [93,94]. Septation is the process whereby the looped heart tube transitions into a multi-chambered heart; this occurs via the addition of cardiomyocytes that contribute to the atrium and the inflow tract [95]. Conditional knockout mice, whereby Tbx5 is haplo-insufficient only in these cardiomyocytes, showed that *tbx5* expression is required for the proliferation of atrial septum progenitors [96]. Furthermore, tissue-specific deletion of *tbx5* in ventricular cardiomyocytes of mice resulted in cardiac contraction dysfunction, which reflects HOS patient symptoms [97]. A *tbx5* mutant strain has been developed in zebrafish, known as the *heartstrings* mutant. In *heartstrings* mutants, the heart tube at 1 dpf is indistinguishable from wild type. Similar to HOS patients and the conditional knockout mice, they display a reduction in rate of contraction. *Heartstrings* hearts do not undergo looping but remain as linear heart tubes. By 3–4 dpf, they stretch to a string-like morphology [94]. Furthermore, double MO knockdowns of both *tbx5* paralogues display defects in cardiac jogging and looping; linear heart tubes remain at the midline or jog to the right, and looping is either abolished or reversed [98].

The role of *tbx5* in eye development has been examined in chick embryos by misexpressing this gene in the ventral retina. This resulted in upregulation of dorsal markers and downregulation of ventral markers, as well as altered projections of retinal ganglion cells [99]. In zebrafish, MO knockdown of both *tbx5* paralogues, *tbx5a* and *tbx5b*, significantly reduces the expression domains of dorsal markers in the eye [98]. Furthermore, transcriptomic analysis revealed that 54 genes were differentially expressed in the eye in these double-morphant embryos [100]. This suggests that *tbx5* is critical in dorsal–ventral patterning of the developing retina in vertebrates, which is consistent with this specific expression observed in *O. alcalica*.

Fish pectoral fins are homologous to tetrapod forelimbs, and genetic loss of function of *tbx5* in mice results in absence of forelimbs [101,102,103]. Similarly, in zebrafish, *tbx5b* morphants develop smaller limbs than controls [98] The LB is established at a specific antero-posterior position of the lateral plate mesoderm. Limb mesenchyme precursors protrude from the trunk to form the LB. At the distal tip of the LB is the apical ectodermal ridge which controls LB growth [104]. *tbx5* is the earliest gene to be expressed in the presumptive pectoral fin mesenchyme, and the regulation of *tbx5* expression in vertebrates has been crucial to understanding the evolutionary origin of paired appendages. In situ hybridisation analysis of *tbx5* during development of the vertebrate sea lamprey (*Petromyzon marinus*), which does not have paired appendages, found *tbx5* expression to be exclusive to the heart field. It was, therefore, hypothesised that *tbx5* expression in the lateral plate mesoderm posterior to the heart is associated with the evolution of pectoral fins [105]. A study aimed at identifying a limb-specific enhancer of *tbx5* identified a potential *cis*-regulatory element downstream from *tbx5* conserved in vertebrates, known as cns12sh. Injection of *tbx5* alongside cns12sh into *heartstrings* embryos, which do not develop pectoral limbs, rescued LB development [105]. However, CRISPR/Cas9 knockout of endogenous cns12sh regulatory elements in mice and zebrafish exhibited normal *tbx5* expression and LB initiation, suggesting there may be other DNA elements that share redundant function or that cns12sh is a pseudo-enhancer that was once required for LB activation but was lost during evolution [106]. Therefore, noncoding DNA elements that regulate *tbx5* expression in the vertebrate LB remain elusive. Further limb enhancers could be identified by comparative analyses of the genomic landscape of vertebrate *tbx5* including *O. alcalica*, which shares this specific expression in the LB.

### 4.3. mef2c

An essential role for Mef2c in heart development was revealed when *Mef2c*-null mice were found to be embryonic lethal due to abnormalities in inflow tract and outflow tract formation, and insufficient cardiac looping results in complete absence of the right ventricle [107,108,109]. More recently, this phenotype was recapitulated in mice with cardiomyocyte-specific *Mef2c* deletions [110]. Similarly, zebrafish Mef2c is crucial in early heart and skeletal muscle development [66,111,112]. Two *mef2c* paralogues exist in zebrafish: *mef2ca* and *mef2cb* [113]. *Mef2cb* is expressed in cardiomyocytes, and MO knockdown of *mef2cb* eliminates cardiomyocytes in the outflow tract, reminiscent of *Mef2c*-null mice [112]. Combinatorial loss-of-function analysis of both paralogues revealed an essential role of *mef2ca* and *b* in controlling the expression of myocardial sarcomeric genes in cardiomyocytes [66].

Although we found no cardiac expression of *O. alcalica mef2c* at the stages we investigated (Figure 4B–G), we conclude that this is because we did not look early enough. In a study of the Mef2 family in Atlantic cod (*Gadus morhua*), *mef2c* was robustly expressed at the cardiac ring stage, during cardiac ring extension at the arterial and venous poles, and after completion of heart formation [114]. Although invertebrates only have a single *mef2* gene, amniotes have four genes (*Mef2a-d*) [115]. The teleost-specific gene duplication has led to six *mef2* genes in zebrafish that undergo extensive alternative splicing [66,116].

We did find extensive expression of *Oa mef2c* in the skeletal muscle cell lineage (in the somites, limb buds, facial muscle, and MMPs) which is also seen in *Xenopus* [67] and zebrafish ([66] (Appendix A). Cells become committed to the myogenic lineage via the cooperation of myogenic regulatory factors (MRFs) Myod, Myf5, Myogenin, and Mrf4, [117] and members of the MEF2 family. *mef2* genes are activated by MRFs and expressed during the terminal differentiation of muscle cells [44,118]. It has been shown that *mef2* genes are required in myogenesis in zebrafish, as *mef2* knockdown by MOs causes downregulation of messenger RNAs (mRNAs) encoding thick filament proteins (myosins), resulting in disrupted sarcomere assembly [111].

Two distinct mechanisms of MMP migration have been described for the body wall and appendage musculature. There is the primitive mechanism of direct epithelial somitic extension, whereby differentiated muscle of the somite extends directly to provide musculature of the body wall. Alternatively, there is a mechanism of long-range migration, whereby MMPs delaminate from somites after undertaking epithelial-to-mesenchymal transition and undergo long-range migration [119]. In zebrafish fin development, some hypaxial muscle precursors are specified in the somites and undergo long-range migration. This was established by assessing expression of *lbx1*, a gene that specifically labels limb muscle precursors [120,121]. In contrast, the body wall musculature of amniotes and the development of chondrichthyan fin muscles are formed via direct extension from the somites [121,122,123]. Therefore, this expression analysis of *mef2c* in *O. alcalica* shows body wall musculature extending from somites, consistent with findings in amniotes [122]. In contrast, it appears that *Oa mef2c*-expressing myocytes in the LB are separated from somites, indicating that long-range migration is the likely mechanism via which muscle precursors move to the LB, similar to terrestrial vertebrates. To conclusively determine the mechanism of hypaxial migration, further expression analyses of genes regulating MMP migration, such as *lbx1* [120,121], should be undertaken.

## 5. Perspectives

Understanding acquired adaptations of *O. alcalica* that allow it to thrive in extreme conditions could be useful when projecting resilience of other freshwater vertebrates living in ecosystems affected by climate change. The developmental stage series presented here reveals *O. alcalica* as a direct-developing teleost and, together with the described methods for undertaking expression analyses, this forms the basis of comparative developmental work with other cichlids. A notable limitation of this study was the difficulty in analysing *O. alcalica* blastula–gastrula stages of development, and, for future studies, it would be valuable to develop a method of observing gene expression in these early embryos.

Overall, our findings show that *O. alcalica* shares the highly conserved molecular regulation of the vertebrate body plan. *gata4* and *tbx5* are expressed in the *O. alcalica* embryonic heart, consistent with these genes having a conserved role in cardiac development in this species. Surprisingly, we also found *Oa gata4* localised to haematopoietic regions of the developing embryo, including yolk sac blood islands, which are characteristic of amniote blood development. The prominence of *Oa gata4* expression in haematopoietic regions sets it apart from other vertebrates; thus, it would be informative to characterise the phylogeny and expression of other members of the GATA family in *O. alcalica*.

## Figures and Tables

**Figure 1 jdb-08-00022-f001:**
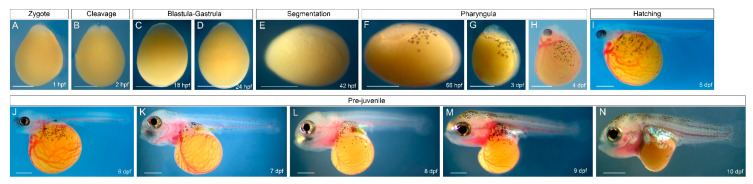
Embryonic and pre-juvenile development of *Oreochromis (Alcolapia)*
*alcalica*. (**A**) At 1 h post fertilisation (hpf), *O. alcalica* embryos were ovoid in shape with the embryonic region located at the animal pole, on top of the large yellow yolk. (**B**) At 2 hpf, the first cell division occurred and two blastomeres were observed. (**C**,**D**) The embryo underwent gastrulation (18 hpf–24 hpf). The cellular dynamics of *O. alcalica* gastrulation were not easily observed due to the large yellow yolk. (**E**) Gastrulation was completed by 42 hpf. The antero-posterior axis could be distinguished, and the neural tube formed. (**F**) At 66 hpf, the antero-posterior axis elongated further. Pigmented melanocytes of the neural crest lineage emerged from the embryonic axis. A clear head and unpigmented eye could be distinguished. (**G**) By 3 days post fertilisation (dpf), there was an increased number of melanocytes migrating on the yolk. The embryo began to vascularise in patches on the yolk and in the area of the beating heart. (**H**) At 4 dpf, the vascular system extended across the yolk and a clearly beating heart was visible with strong blood circulation. The eye developed pigmentation. (**A**–**H**) Throughout these stages, the embryo was surrounded by the chorion membrane. (**I**) At 5 dpf, embryos hatched from the chorion. (**J**–**N**) Following hatching at 6–10 dpf, the yolk decreased in size and embryos rapidly developed adult morphology. (**A**–**D**) Animal pole to the top. (**E**,**F**,**I**–**N**) Lateral views; anterior to the left. (**G**,**H**) Lateral views; anterior to the top. Scale bars correspond to 1 mm. Heart contraction was observed and recorded at 66 hpf, 3 dpf, 4 dpf, and 5 dpf (see Appendix A).

**Figure 2 jdb-08-00022-f002:**
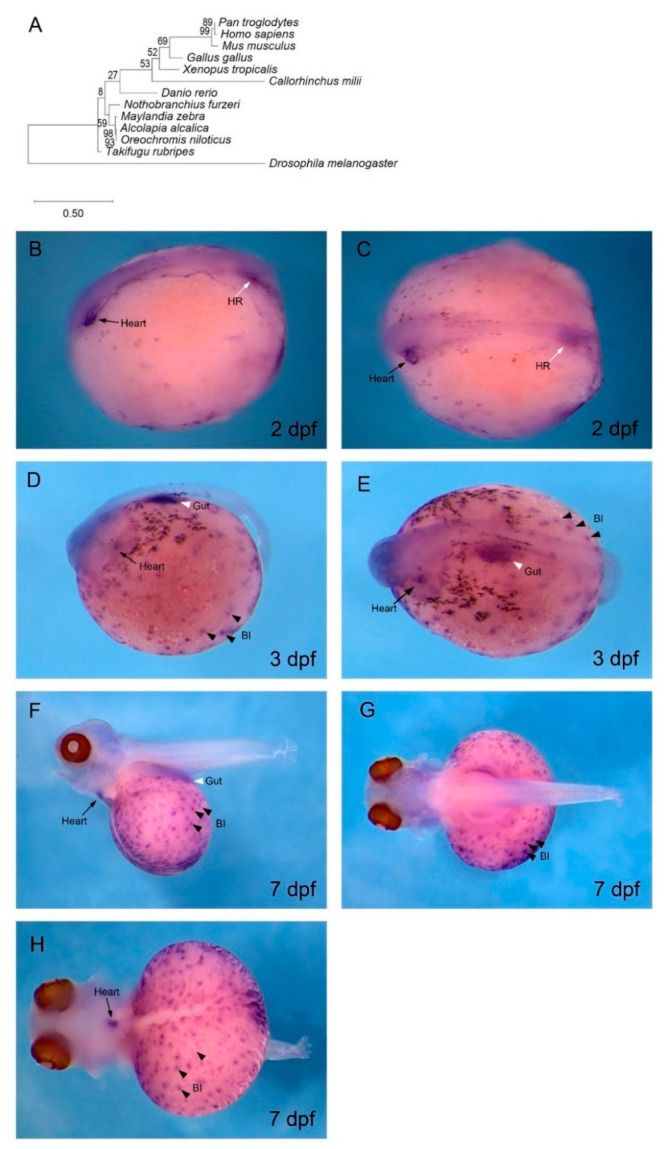
*gata4* is expressed in cardiac and vascular regions of *O. alcalica* embryos. (**A**) Phylogenetic analysis of GATA4. The evolutionary history was inferred using the maximum likelihood method and Dayhoff with frequency model and gamma distribution [58]. (**B**–**H**) In situ hybridisation of *O. alcalica gata4* at 2 days post fertilisation (dpf) (**B**,**C**), 3 dpf (**D**,**E**), and 7 dpf (**F**–**H**). At 2 dpf, *Oa gata4* was expressed in the heart and haematopoietic region (HR) (**B**,**C**). At 3 dpf and 7 dpf, *Oa gata4* was expressed in the heart, gut, and blood islands (BI) (**D**–**H**). At 7 dpf, embryos were bleached to remove melanocyte pigmentation (**F**–**H**). Black arrows denote the heart, black arrowheads mark blood islands (BI), white arrows denote the haematopoietic region (HR), and white arrowheads mark the gut. (**B**,**D**,**F**) Lateral views; anterior to the left. (**C**,**E**,**G**) Dorsal views; anterior to the left. (**H**) Anterior view; dorsal to the left.

**Figure 3 jdb-08-00022-f003:**
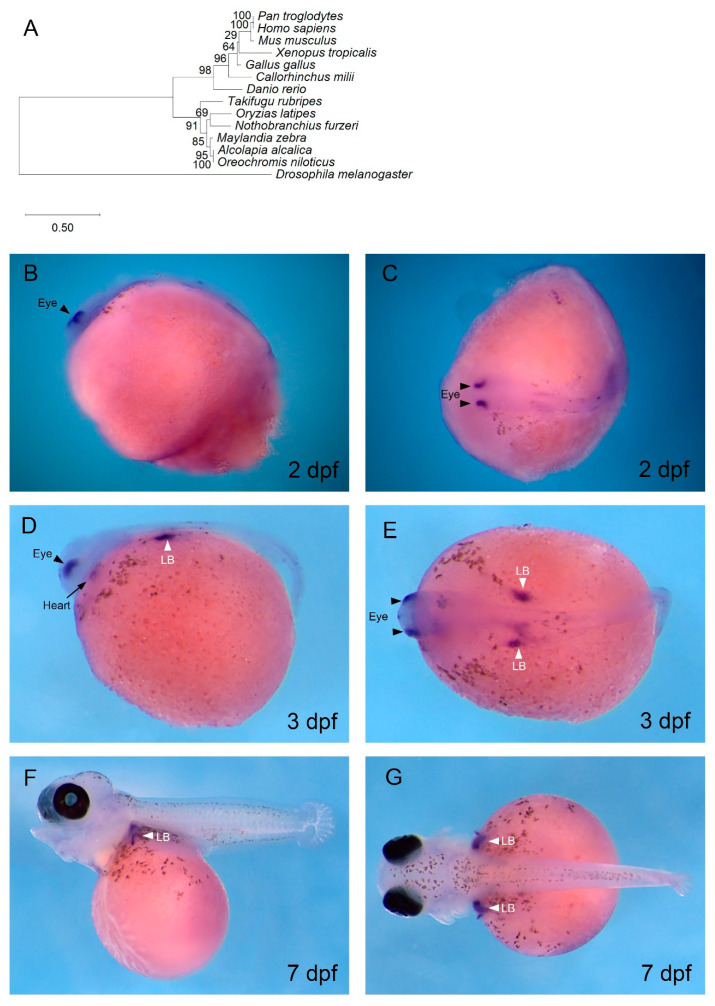
*tbx5* expression in *O. alcalica* dorsal retina, heart, and pectoral limb buds. (**A**) Phylogenetic analysis of Tbx5. The evolutionary history was inferred using the maximum likelihood method and Jones with frequency model [64]. (**B**–**G**) In situ hybridisation of *O. alcalica tbx5* at 2 days post fertilisation (dpf) (**B**,**C**), 3 dpf (**D**,**E**), and 7 dpf (**F**,**G**). At 2 dpf, *tbx5* was expressed in the dorsal region of the eye (**B**,**C**). At 3 dpf, *tbx5* was expressed in the eye, heart, and limb bud (LB) (**D**,**E**). At 7 dpf, *tbx5* was expressed in the LB (**F**,**G**). Black arrows denote the heart, black arrowheads mark the eyes, and white arrowheads denote the limb buds (LB). (**B**,**D**,**F**) Lateral views; anterior to the left. (**C**,**E**,**G**) Dorsal views; anterior to the left.

**Figure 4 jdb-08-00022-f004:**
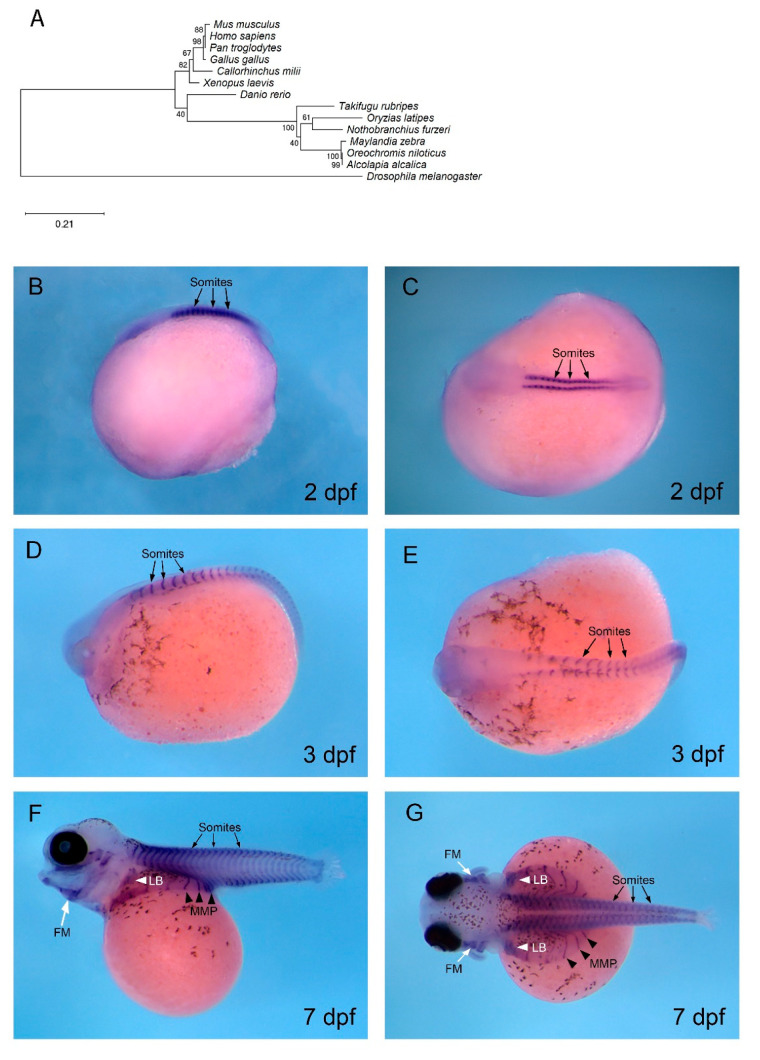
*mef2c* is expressed in developing muscle in *O. alcalica*. (**A**) Phylogenetic analysis of Mef2c. The evolutionary history was inferred by the maximum likelihood method and Jones with frequency model [64]. (**B**–**G**) In situ hybridisation of *O. alcalica mef2c* at 2 days post fertilisation (dpf) (**B**,**C**), 3 dpf (**D**,**E**), and 7 dpf (**F**,**G**). At 2 dpf and 3 dpf, *mef2c* was expressed in the somites (**B**–**E**). At 7 dpf, *mef2c* was expressed in somites, limb buds (LB), migrating muscle precursors (MMP), and facial muscle (FM). Black arrows denote the somites, black arrowheads mark migrating muscle precursors (MMP), white arrows denote facial muscle (FM), and white arrowheads mark limb buds (LB). (**B**,**D**,**F**) Lateral views; anterior to the left. (**C**,**E**,**G**) Dorsal views; anterior to the left.

**Table 1 jdb-08-00022-t001:** *Oreochromis (Alcolapia) alcalica* and zebrafish primers for RT-PCR and probe synthesis.

Gene	Forward Primer 5′–3′	Reverse Primer 5′–3′
*O. alcalica gata4*	TGTCTCCGCGCTTCACCTTCTCCA	GACCGGCTCTCCCTCTGCGTTCC
*O. alcalica mef2c*	ATGGGGCGAAAGAAGAT	AGCCCACCCTGATTACTG
*O. alcalica tbx5*	AGAGGCAGCGACGACAATGAGC	GGGGGATAGGAGGAGGGGTGATAG
Zebrafish *gata4*	CCTACAGGCACCCCAGCAGAGCAG	CCCGCCGCCACAGAGGAGTC
Zebrafish *mef2ca*	TTGCGCGGATAATGGACGAACG	GGGGGCCGGTGGGTGACTC
Zebrafish *tbx5b*	GCTCCCCGCCACTACAAACTCAG	GATATGTCCGAAAGGGTCCAGGTG

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
