# Peer review of "Exploring the Expression of Cardiac Regulators in a Vertebrate Extremophile: The Cichlid Fish *Oreochromis (Alcolapia) alcalica"

_jdb, 2020, doi:10.3390/jdb8040022_

Round 1
Reviewer 1 Report
The study presents the expression of cardiac regulators in developing of cichlid fish Oreochromis (Alcolapia) alcalica. My objections see below:
Title: Authors should remove “adapted to an extreme environment” - because current research is devoted to expression of some genes in Oreochromis alcalica embrions
28: Authors wrote «the adaptation to extreme conditions» - but according to the obtained data there is no adaptation to extreme conditions. Please, remove this phrase from the Abstract – to avoid confusion.
27-28: This phrase does not apply in any way to the work and the conclusions made by the authors. This phase should to be moved to the «Perspectives» section.
94-96 The introduction should be devoted to the literature data on the studied problem. You may provide your data at the end of the introduction. Please, delete or move this phrase to the end for the Abstract section
“Results” section
Global note. Authors discussed obtained data with literature. Please, move all “discussions” to Discussion section.
The authors provide developing data (Figure 1) only on the light microscopy level with low magnification. These data do not allow to verify the development of Oreochromis alcalica.
197-200: This phase is not related with current research. Please, delete this phrase or move it to the Discussion section
204 – I did not see «neural tube» - please, provide photo with high resolution and magnification
215-218: There is no place for your future plans in the Results section – please, transfer to the last paragraph of Discussion section or in Perspective section
220: According to this phase (without a prolonged, free-feeding larval stage) Oreochromis alcalica has short-living larva. Please, rephrase this sentence.
219-227: Move this phrase to Discussion section.
228-234. It is impossible to compare the development of embryos by the hours, because it is known that the main factor in development is physical factors (for example, temperature).
228-241: Move this phrase to Discussion section
273-277 : Move this phrase to Discussion section
306-309: Move this phrase to Discussion section
337-354: Move this phrase to Discussion section
Discussion section
401: I could not understand what namely “is interesting” – please, rephrase this sentence.
415-417: Please, set reference for this sentence
“Perspectives” section
The data obtained by the authors indicate evolutionary conservatism of “gata4, tbx5 and mef2c” genes of O. alcalica. In is not clear what the authors meant «evolutionary adaptations of O. alcalica to extreme conditions». Please, clarify this phrase.
Author Response
Reviewer 1:
Title: Authors should remove “adapted to an extreme environment” - because current research is devoted to expression of some genes in Oreochromis alcalica embrions
changed title
28: Authors wrote «the adaptation to extreme conditions» - but according to the obtained data there is no adaptation to extreme conditions. Please, remove this phrase from the Abstract – to avoid confusion.
Changed wording
27-28: This phrase does not apply in any way to the work and the conclusions made by the authors. This phase should to be moved to the «Perspectives» section.
It points to the importance of the work and provides a useful narrative pointing to potential impact
94-96 The introduction should be devoted to the literature data on the studied problem. You may provide your data at the end of the introduction. Please, delete or move this phrase to the end for the Abstract section
it is very common. and widely accepted (even expected) for authors to include a brief summing-up of what the paper shows at the end of the intro
“Results” section
Global note. Authors discussed obtained data with literature. Please, move all “discussions” to Discussion section.
This was a conscious style decision; we provide enough of a narrative around the results to maintain reader interest and to provide the context of the experimental work
The authors provide developing data (Figure 1) only on the light microscopy level with low magnification. These data do not allow to verify the development of Oreochromis alcalica.
I disagree. This are very well photographed images of a stage series of embryos during the first ten days of development providing a dramatic visual on how different these embryos look from the very well-studied zebrafish
197-200: This phase is not related with current research. Please, delete this phrase or move it to the Discussion section
this has been removed as suggested
204 – I did not see «neural tube» - please, provide photo with high resolution and magnification
Fair point, reference to the neural tube has been deleted
215-218: There is no place for your future plans in the Results section – please, transfer to the last paragraph of Discussion section or in Perspective section
This has been removed as suggested
220: According to this phase (without a prolonged, free-feeding larval stage) Oreochromis alcalica has short-living larva. Please, rephrase this sentence.
This sentence has been edited
219-227: Move this phrase to Discussion section.
This has been removed as suggested
228-234. It is impossible to compare the development of embryos by the hours, because it is known that the main factor in development is physical factors (for example, temperature).
We now include the temperature (30C)
228-241: Move this phrase to Discussion section
we include this narrative around the results to provide the context of the experimental work
273-277 : Move this phrase to Discussion section
we include this narrative around these results highlight significance of this particular finding and maintain reader interest
306-309: Move this phrase to Discussion section
This sentence refers directly to data shown in supplementary material (ie, the zebrafish expression data)
337-354: Move this phrase to Discussion section
This reference to previous studies provides context for our surprising result ;
Discussion section
401: I could not understand what namely “is interesting” – please, rephrase this sentence.
The sentence needed some punctuation; this has made it more clear
415-417: Please, set reference for this sentence
Apologies, this reference was omitted and should be included:
Ueno H, Weissman IL. The origin and fate of yolk sac hematopoiesis: application of chimera analyses to developmental studies. Int J Dev Biol. 2010;54(6-7):1019-31. doi: 10.1387/ijdb.093039hu.
“Perspectives” section
The data obtained by the authors indicate evolutionary conservatism of “gata4, tbx5 and mef2c” genes of O. alcalica. In is not clear what the authors meant «evolutionary adaptations of O. alcalica to extreme conditions». Please, clarify this phrase.
We have re-phrased this to ‘ acquired adaptations of O. alcalica that allow it to thrive in extreme conditions’
Reviewer 2 Report
In the study, the authors examined the heart development of O. a. The most important genes expressed during heart development have been analysed via in situ hybridization. The study is very interesting and well described.
My biggest concern is the missing graphs/pictures of the RT-PCRs. The authors described this method and used the PCR products for in situ hybridization. Therefore, it would be nice to add the graphs in this present manuscript - as supplements. Additionally, I would like to know if there is another paper planned with RT-qPCR - this would be great – I would look forward to it (How is the gene expression during the development of O.a. Is there also a difference to ZF or even trout noticeable?)
some minor corrections
-Citation for fish number is missing (Line 56), maybe cite Betancur et al. 2017 I thought there are around 34.000 species? Please check.
-Some smaller correction, like missing blank spaces (line 177)
-figures of the family trees are a bit hard to read – enlarge
Perspective: As there are differences in the GATA4 expression between O.a. and ZF, I would like to read a perspective more in this direction. What is the future aim? Is ZF maybe not suitable to be a model organism for “all” fishes?
It was very nice to review it :)
Author Response
Reviewer 2
In the study, the authors examined the heart development of O. a. The most important genes expressed during heart development have been analysed via in situ hybridization. The study is very interesting and well described.
My biggest concern is the missing graphs/pictures of the RT-PCRs. The authors described this method and used the PCR products for in situ hybridization. Therefore, it would be nice to add the graphs in this present manuscript - as supplements. Additionally, I would like to know if there is another paper planned with RT-qPCR - this would be great – I would look forward to it (How is the gene expression during the development of O.a. Is there also a difference to ZF or even trout noticeable?)
We have not undertaken an expression analysis using rtPCR or qPCR (although this is a good suggestion for future work); we simply used RT-PCR to generate the plasmid in situ probe. We could include the pictures of these PCR products on a gel, but I don’t think it would be very informative.
some minor corrections
-Citation for fish number is missing (Line 56), maybe cite Betancur et al. 2017 I thought there are around 34.000 species? Please check.
We now include a reference for teleost numbers
-Some smaller correction, like missing blank spaces (line 177)
corrected
-figures of the family trees are a bit hard to read – enlarge
making the phylogeny panel bigger will reduce the size of the in situs; I would rather not but can do if required.
Perspective: As there are differences in the GATA4 expression between Oa. and ZF, I would like to read a perspective more in this direction. What is the future aim? Is ZF maybe not suitable to be a model organism for “all” fishes?
Yes, that is a clear point from this work; we hope we have made that clear now.
It was very nice to review it :)
Thanks (Gemma was a v good student!)
Reviewer 3 Report
In their paper, Sutton et al. describe the different morphological features of the non-model species Alcolapia alcalica during embryonic development and investigate the expression of 3 genes orthologous to gata4, tbx5 and mef2c. This is a nice and interesting study, and the paper in very well written. The article fits well to the scope of J of dev Bio. Here are several remarks and suggestions:
- The main criticism I have is on the de facto assumption that the sequences cloned from O. alcalica correspond to the true tbx5, mef2c and gata4 orthologues. The authors should be more cautious and discuss the possibility of false assignment, as they rely on partial sequences of < 500 bp. This point is particularly important but not specific to Mef2. The different Mef2 paralogues are relatively distant in zebrafish but some part are very well conserved (% identity is 81% over 300 bp between mef2ca and mef2d). Is it possible that the portion of the O alcalica sequenced used for the in situ hybridisation falls in such highly conserved mef2 domain? I suggest to discuss this properly in the paper and to make the required adjustment of phrasing in the results for the 3 genes they partially cloned.
- I understand that the oligonucleotides are derived from an unpublished de novo assembly of O alcalica. Please indicate an author name as unpublished results in order to point the reader to the lab owning these data.
- The authors should deposite in genbank the O alcalica sequences obtained in this study. It is important for the community to be able to access such non-model species data.
- The picture acquisition for the WISH with a blue background is hiding the purple signal of the DIG Ap labelling. Modifying the setup to have a white background (as it is usually the case for such data) would be much better.
- In the material and methods the line from 134 to 143 should be in a new separated paragraph of the material and Methods. In addition parts of the legend 2A, 3A and 4A describing how the dendrogram was draw are clearly methodology and should be moved to the material and methods.
- The authors should produce a supplementary table with the accession number (genbank or else) of the orthologues used to draw the dendrogram.
- In the results describing the dendrogram the authors should show the % id obtained for the different O alcalica orthologues against the zebrafish gene.
Author Response
Reviewer 3
In their paper, Sutton et al. describe the different morphological features of the non-model species Alcolapia alcalica during embryonic development and investigate the expression of 3 genes orthologous to gata4, tbx5 and mef2c. This is a nice and interesting study, and the paper in very well written. The article fits well to the scope of J of dev Bio. Here are several remarks and suggestions:
The main criticism I have is on the de facto assumption that the sequences cloned from O. alcalica correspond to the true tbx5, mef2c and gata4 orthologues. The authors should be more cautious and discuss the possibility of false assignment, as they rely on partial sequences of < 500 bp. This point is particularly important but not specific to Mef2. The different Mef2 paralogues are relatively distant in zebrafish but some part are very well conserved (% identity is 81% over 300 bp between mef2ca and mef2d). Is it possible that the portion of the O alcalica sequenced used for the in situ hybridisation falls in such highly conserved mef2 domain? I suggest to discuss this properly in the paper and to make the required adjustment of phrasing in the results for the 3 genes they partially cloned.
Even though the ISH probes were made from partial cDNAs, all alignments were done using full length protein sequences; the full length Alcolapia sequences have been deposited in genbank.
this is now clearly stated in the text
in addition we now present another MEF2 Alignment including MEF2D orthologues shows that our Alcolapia mef2c clades with other vertebrate mef2c and not mef2d
I understand that the oligonucleotides are derived from an unpublished de novo assembly of O alcalica. Please indicate an author name as unpublished results in order to point the reader to the lab owning these data.
This is now included (Kanchon Dasmahapatra)
The authors should deposite in genbank the O alcalica sequences obtained in this study. It is important for the community to be able to access such non-model species data.
Done; line 152
The picture acquisition for the WISH with a blue background is hiding the purple signal of the DIG Ap labelling. Modifying the setup to have a white background (as it is usually the case for such data) would be much better.
The photography is very clear; these yolky embryos are more similar to Xenopus (in terms of refractile nature and large size) than they are to zebrafish and in our experience are imaged better with a blue background (as is standard for Xenopus)
In the material and methods the line from 134 to 143 should be in a new separated paragraph of the material and Methods.
Formatting error
In addition parts of the legend 2A, 3A and 4A describing how the dendrogram was draw are clearly methodology and should be moved to the material and methods.
This has been moved to the materials and methods
The authors should produce a supplementary table with the accession number (genbank or else) of the orthologues used to draw the dendrogram.
The sequences were submitted to Genbank: Oa 152 tbx5 (MT904199), Oa mef2c (MT904200), Oa gata4 (MT904201).
In the results describing the dendrogram the authors should show the % id obtained for the different O alcalica orthologues against the zebrafish gene.
Alcolapia and zebrafish protein identity at level of amino acid sequence: 60% identity tbx5; 63% identity mef2c; 62% identity gata4; this is now included in methods
Round 2
Reviewer 1 Report
Despite the fact that the authors have corrected most of my comments, nevertheless, my main remark regarding the interpretation of the stages of embryonic development still remains.
Major
In current research the Authors have used a dissecting microscope - which does not allow tracing the early stages of development. At the same time, the authors wrought about this in Line 208-216. I strongly recommend to exclude early development from the Results. So, only lines 244-258 from the “Early development of O. alcalica” section should be stayed. Otherwise - Authors must provide the evidences of each early development stages (cleavage, blastula stage, gastrulation, segmentation). My comments about this see below.
Line 249-251:This proposition is not supported by any factual material or literature data. Authors should expand this sentence or delete it.
In line 218-219 Authors wrought "embryos have completed gastrulation" and “entered segmentation period” and reffered to Figure 1E. But it is impossible to recognize which stage of embryonic development of O. alcalica
Line 218-219 It is not obvious how the pharyngula period differs from the segmentation period. It would be nice if the authors describe the main features of each of the periods.
Line 201-202 – On the Figures 1 A-H I did not see chorion – please, point to the chorion on these images
Fig.1 C, D – Authors wrought “the embryo undergoes gastrulation” – But on the Figure1 C and D I cannot understand is it “cleavage” or “blastula” or “gastrula”.
Author Response
Despite the fact that the authors have corrected most of my comments, nevertheless, my main remark regarding the interpretation of the stages of embryonic development still remains.
Sorry for missing this; we now change our interpretation as recommended and have deleted much of the text..
Major
In current research the Authors have used a dissecting microscope - which does not allow tracing the early stages of development.
The early stages are difficult to visualise in detail; we now change and remove text as suggested.
At the same time, the authors wrought about this in Line 208-216. I strongly recommend to exclude early development from the Results.
We have now deleted most of this section as recommended; we include a modified description of Figure 1, that takes into account the limitations of the microscopy used.
So, only lines 244-258 from the “Early development of O. alcalica” section should be stayed. Otherwise - Authors must provide the evidences of each early development stages (cleavage, blastula stage, gastrulation, segmentation). My comments about this see below.
Line 249-251:This proposition is not supported by any factual material or literature data. Authors should expand this sentence or delete it.
This section has been deleted and our description of early development is now toned down to fit the limited data we show.
In line 218-219 Authors wrought "embryos have completed gastrulation" and “entered segmentation period” and reffered to Figure 1E. But it is impossible to recognize which stage of embryonic development of O. alcalica
We have changed the text associated with Figure 1E.
Line 218-219 It is not obvious how the pharyngula period differs from the segmentation period. It would be nice if the authors describe the main features of each of the periods.
We now describe the pharyngula as the phylotypic stage with some comments on its significance.
Line 201-202 – On the Figures 1 A-H I did not see chorion – please, point to the chorion on these images
We include an additional supplemental figure with enlarged version of Figure 1A where the chorion is visible and a labelled Figure 1E
Fig.1 C, D – Authors wrought “the embryo undergoes gastrulation” – But on the Figure1 C and D I cannot understand is it “cleavage” or “blastula” or “gastrula”.
This has been removed